# Development and Application of Digital Twin–BIM Technology for Bridge Management

**Elfrido Elias Tita** [1] , **Gakuho Watanabe** [1,*] , **Peilun Shao** [1] and **Kenji Arii** [2]

[1] Department of Civil and Environmental Engineering, Yamaguchi University, 2-16-1, Tokiwadai, Ube City, Yamaguchi 7558611, Japan; neuz2012@gmail.com (E.E.T.); a504wdu@yamaguchi-u.ac.jp (P.S.)

[2] Structural and Bridge Engineering Division, Chodai Co., Ltd., 17-18 Teppo-cho, Naka-ku, Hiroshima City, Hiroshima 7300017, Japan; arii-k@chodai.co.jp

* Correspondence: gakuho.w@yamaguchi-u.ac.jp

**Abstract:** The concept and technology of a digital twin, which represent a replica of a real object in a virtual space called Industry 4.0, are widely used across all industries and purposes. Similarly, in the architecture, engineering, and construction (AEC) industries, there is an urgent need to develop a technology called BIM, a form of digital twin based on 3D models, for the purpose of improving productivity and reducing costs. Bridge structures are required to be safe, reliable, and durable, and various research studies have been conducted on maintenance and repair strategies and their development by fusing health monitoring and digital twins. In this study, we explore the development of digital twin–BIM technology and demonstrate its various applications for an existing bridge structure where the implementation of health monitoring is planned. Moreover, we evaluate the characteristics of the structural performance of the bridge structure using digital twin–BIM technology.

**Keywords:** digital twin; BIM; landscape visualization; traffic simulation; FE analysis; maintenance





## 1. Introduction

A digital twin (DT) refers to a virtual replication of a physical object, system, or process, serving as a valuable tool across diverse industries for several purposes, such as product development, manufacturing and operations, asset management, smart cities and infrastructures, healthcare, and energy and utilities. Digital twin technology has been a trending technology in recent years in all industries. The importance of the revolutionary technologies called Construction 4.0 is extremely increasing in the industry [1,2].

A specific form of the digital twin concept widely adopted in the architecture, engineering, and construction (AEC) industry is building information Modeling (BIM), which facilitates the creation of virtual representations of structures such as buildings, bridges, roads, and other infrastructure components. The employment of BIM models offers numerous advantages and benefits various stakeholders involved in the construction and building life cycle. By simulating and analyzing multiple design alternatives, quantities, and construction sequences, BIM empowers the resource allocation optimization, the identification of potential cost-saving measures, and the enhancement of project scheduling. These capabilities translate into reduced material waste, improved construction sequencing, and an enhanced overall project efficiency, leading to significant time and cost savings [3–6].

Bridges play a critical role in modern transportation infrastructure, facilitating the movement of people, goods, and services. Ensuring their safety, reliability, and durability are of paramount importance to the communities they serve. However, bridge management is a complex and challenging task, requiring careful planning, structural health monitoring (SHM), and maintenance [7,8]. Additionally, one of the most interesting advantages of digital twin is that it can monitor, visualize, and evaluate the time-variable performance of the targeted structures by way of integrating ICT, IoT, and various information technologies.

From the above backgrounds, many studies have been conducted on the development of structural health monitoring systems based on digital twins and their implementation for the maintenance and management of infrastructure facilities [9–20].

Our research group conducted research on structural performance evaluation and anomaly detection based on monitoring long-span bridges using GNSS satellites and other technologies. Through this research, it was found that the structural analysis utilizing digital twin–BIM technology is extremely essential for verifying the structural performance of existing bridge structures based on GNSS monitoring and sensors [21,22]. In addition, a research project on maintenance strategies based on the GNSS structural health monitoring for the targeted bridge in this study (Comoro Bridge) is planned to start in Timor Leste. Therefore, in this study, we explore the application of digital twin–BIM technology for bridge management and investigate the fundamental characteristics of structural performance based on an FE analysis using digital twin–BIM technology.

## 2. PC Box-Girder Bridge of Comoro

The Comoro Bridge is a prestressed concrete box-girder bridge constructed in Dili, East Timor, as shown in Figures 1–3. Dili is East Timor's capital as well as its administrative and economic center. Since there was only one bridge connecting West Dili and East Dili prior to the construction of this bridge, traffic congestion was very serious during that period. From this reason, there was a strong demand for the construction of a transportation system that could address the heavy traffic, and finally, the construction of a new bridge was promoted.

The construction of the bridge began in August 2016 and the bridge was opened to the public in September 2018. It spans a total length of 250 m with a width of 11.50 m. It consists of six spans, including two abutments at both side span and five piers (35 m + 4 at 45 m + 35 m). Positioned over the Comoro River, it connects the western and eastern parts of Dili. Since it played a significant role as a transportation facility in improving traffic services within the City of Dili, structural health monitoring was planned to be conducted. Therefore, it was chosen as a target bridge for this study.

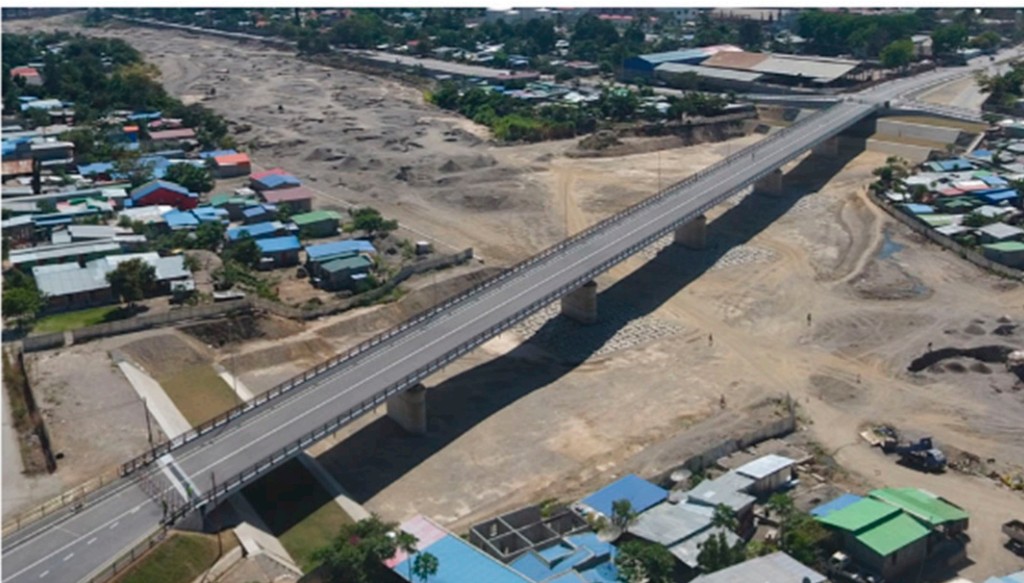

**Figure 1.** Existing Comoro Bridge.

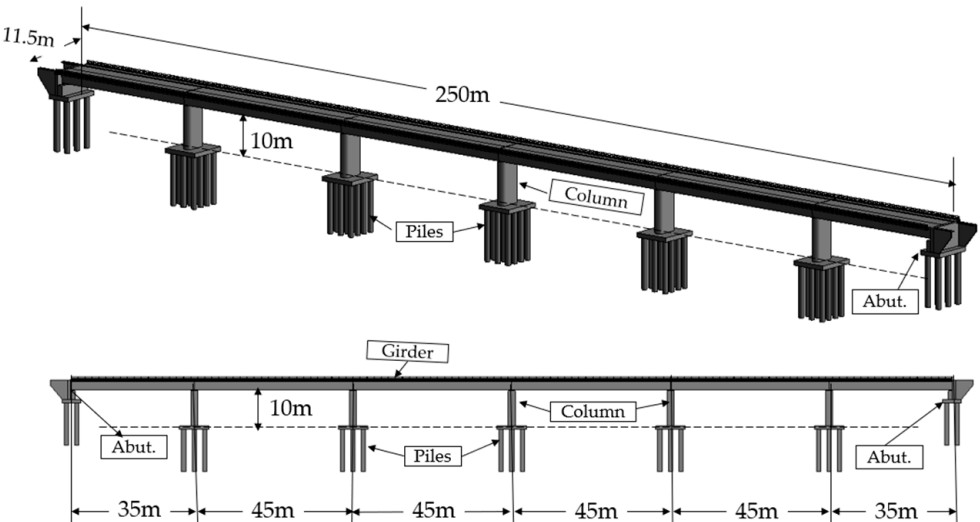

**Figure 2.** Geometrical model of Comoro Bridge.

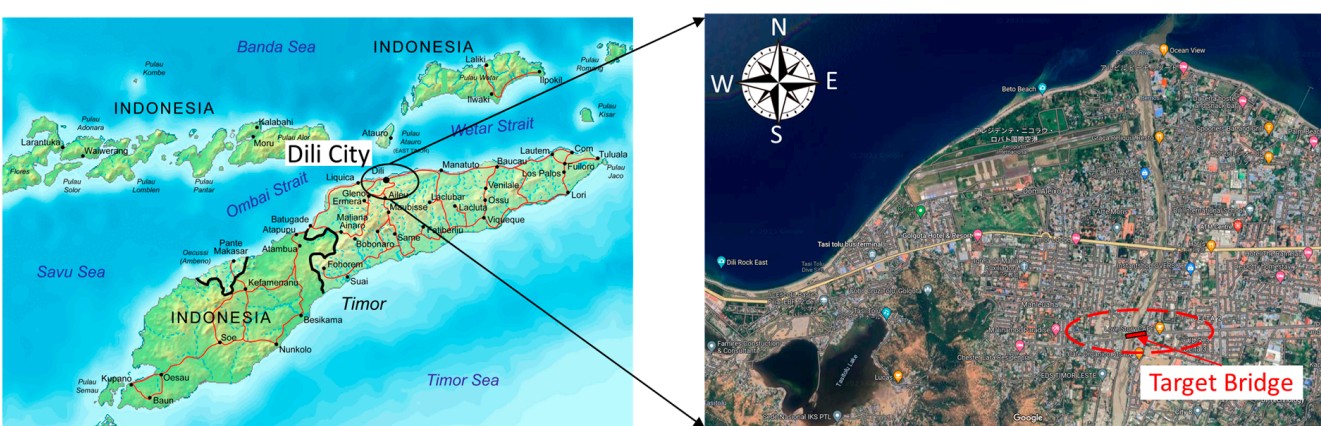

**Figure 3.** Location map.

### 3. Digital Twin and BIM Strategy

*3.1. Digital Twin*

A digital twin (DT) is a highly promising technology as it enables to realize smart manufacturing and Industry 4.0. It is characterized by the seamless integration between cyber and physical spaces [1,2]. Moreover, digital twin technology provides a dynamic and up-to-date representation of a physical object or system. By consolidating all data in one centralized location, the digital twin evolves in real time with inputs from sensors and other sources. It offers several benefits, including: (a) An optimized performance with actionable insights, simulations, and a holistic view of both online and offline data enabling swift and informed decision-making to optimize performance and efficiency. (b) A comprehensive data visibility: by eliminating data silos and integrating historical and real-time data, the digital twin unlocks value throughout the life cycle of a project. (c) Intelligent predictions: real-time sensor data and predictive recommendations through machine learning and artificial intelligence greatly enhance maintenance and operational capabilities. (d) Faster results: creating digital twins of complex assets, factories, and processes exponentially enhances operational value, reduces development efforts, and accelerates time to market. Therefore, in this study, we explored the use of the digital twin–BIM technology for landscape visualization, traffic flow simulation, finite element analysis, and structural maintenance.

### 3.2. Building Information Modeling (BIM)

Building information modeling (BIM) has become an invaluable tool for architects, engineers, and the construction industry. The U.S. National Building Information Model Standard Project Committee has defined BIM as a digital representation of a facility's physical and functional characteristics. BIM facilitates improved collaboration among architects, engineers, contractors, and other project participants. It provides a shared digital platform where all relevant project information can be accessed, updated, and shared in real time, fostering better communication and coordination. BIM data are utilized during the entire process of a structure's life cycle, including its design, construction, maintenance, and renovation, as depicted in Figure 4. By facilitating an improved communication and coordination among team members, BIM minimizes errors, reduces rework, and enhances efficiency in the design, construction, and operation of buildings and infrastructure projects. Consequently, BIM offers several advantages, including maximizing efficiency, reducing cost, reducing waste, improved cost estimates, enhanced project insights, effective communication and collaboration, risk mitigation, and superior outcomes.

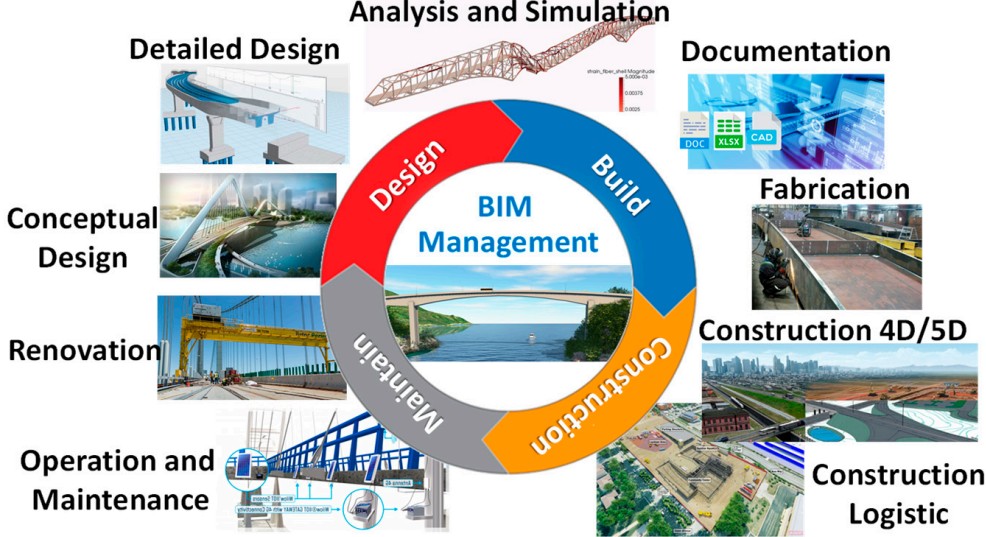

**Figure 4.** BIM process.

### 3.3. Development of a Digital Twin–BIM Model for Bridge Management Workflows

The development of the digital twin–BIM model in this research is presented in Figure 5. This workflow is designed to accommodate bridges of any shape. Specifically, this study focuses on modeling a PC box-girder bridge in Dili City, Timor-Leste. The modeling processes are as follows: (1) Data sources: in order to accurately idealize the existing bridge, it is essential to gather bridge information and documents, such as the design report and construction drawings, from the owner or relevant institution. (2) BIM design tool (Revit): The substructure and superstructure components of the bridge are created as BIM objects (Revit families), as illustrated in Figure 6. Once the components are developed, they are loaded into the BIM template (Revit project) to facilitate model integration. Ultimately, the digital-twin bridge model is established, as shown in Figure 7. (3) Additionally, a visual programming language (Dynamo) is utilized, as depicted in Figure 8. Dynamo allows access to the Revit API, enhancing productivity for designers and modelers. (4) Upon completion of the digital twin model, it can be integrated with BIM simulation tools for landscape visualization and traffic flow simulation. Furthermore, the digital model can be linked with a finite element (FE) analysis to evaluate structural performance. Lastly, the model can be employed for structural maintenance purposes.

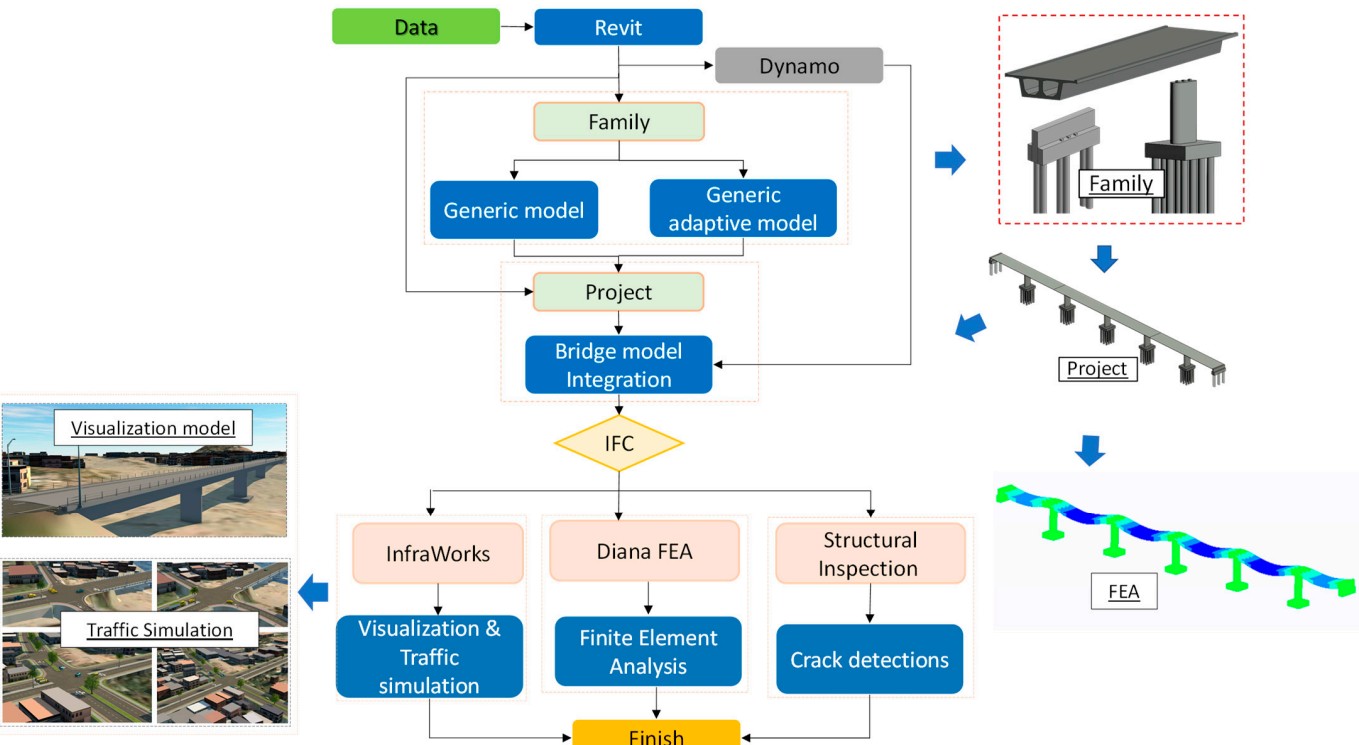

**Figure 5.** The Development of a Digital Twin–BIM Model for Bridge Management Workflow.

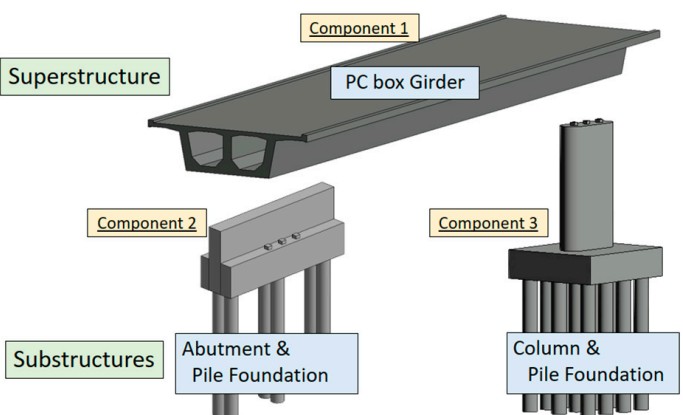

**Figure 6.** Bridge components.

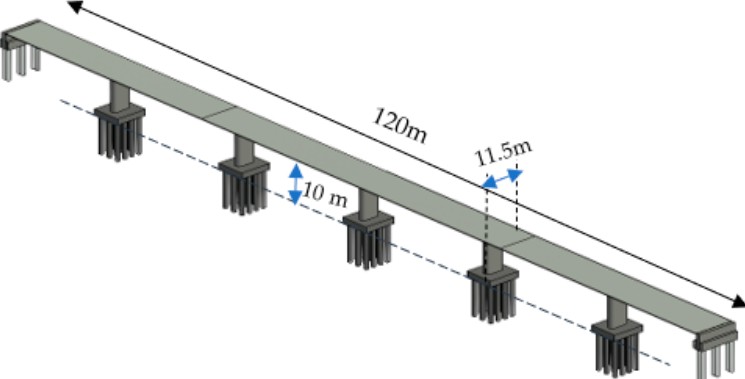

**Figure 7.** Bridge model.

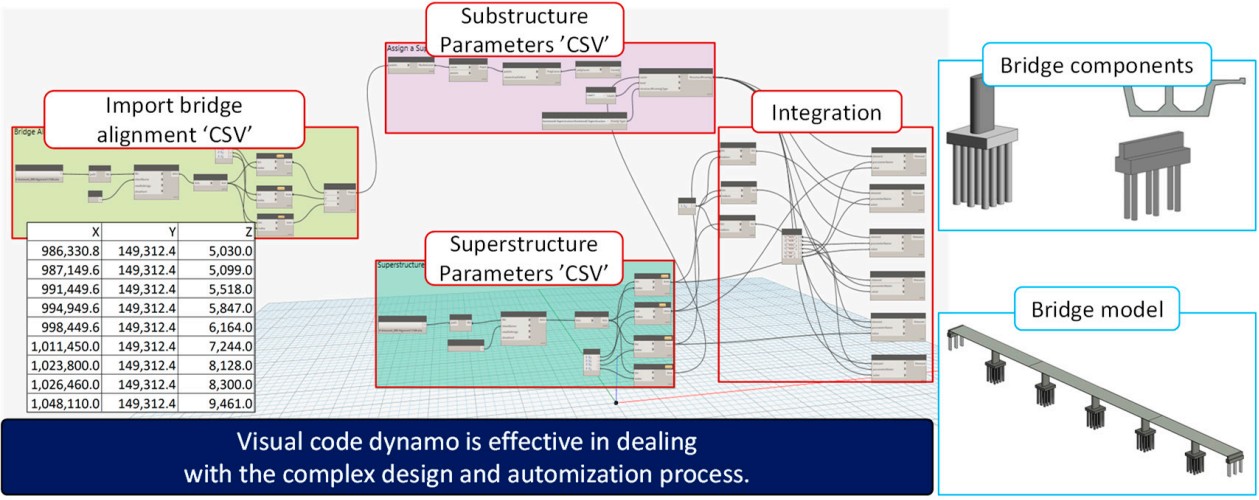

**Figure 8.** Dynamo Visual Programming Language.

## 4. Building Information Management

### 4.1. BIM Management Applications Using Digital Twin of Comoro Bridge

In this study, BIM management was implemented through the utilization of the digital twin of Comoro Bridge for landscape visualization, traffic management, structural performance evaluation, and structural maintenance, as illustrated in Figure 9.

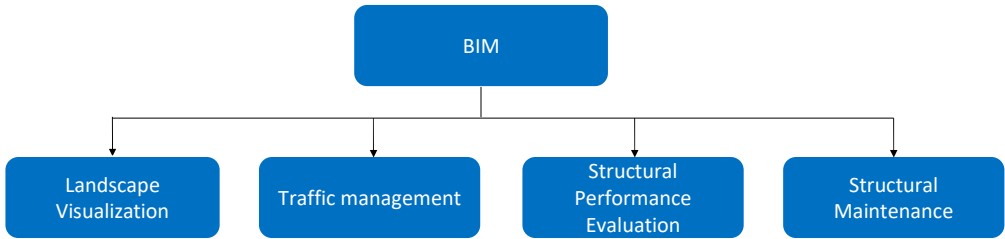

**Figure 9.** BIM management applications.

### 4.2. Digital Twin of Comoro Bridge for Landscape Visualization

By utilizing Building information modeling (BIM) technology, the physical model of the Comoro Bridge can be effortlessly replicated into cyberspace, as shown in Figures 10 and 11. This enables the digital twin model to comprehensively visualize the Comoro Bridge encompassing its actual surrounding environment.

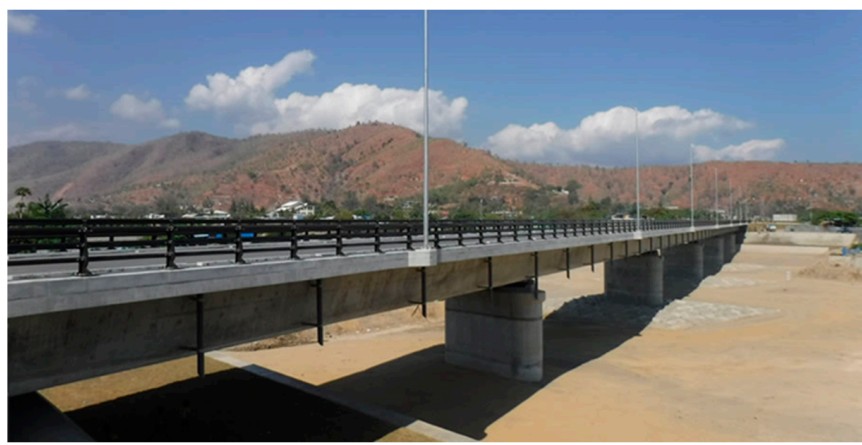

**Figure 10.** Real bridge model.

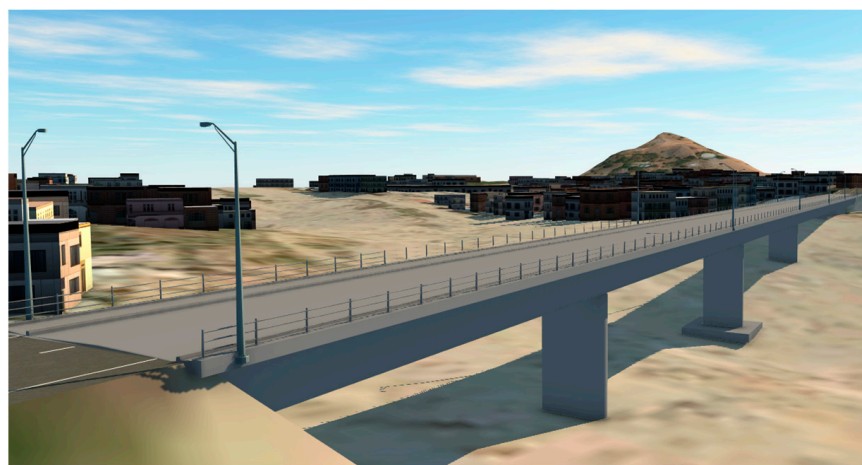

**Figure 11.** Digital twin model.

### 4.3. Digital Twin of Comoro Bridge for Traffic Flows Simulation

The Comoro Bridge connects to the existing road across the Comoro River as depicted in Figure 12a. Dili's main arterial roads are connected to multiple roads as depicted in Figure 12b, and they are forming a road traffic network. The development of the digital twin–BIM model of Dili's road network, specifying the road inflows and outflows from each road (Figure 12c) and traffic events (Figure 12d), enables the traffic flow simulation on existing road networks as depicted in Figure 12e. Such a traffic flow simulation was performed to verify the following objectives.

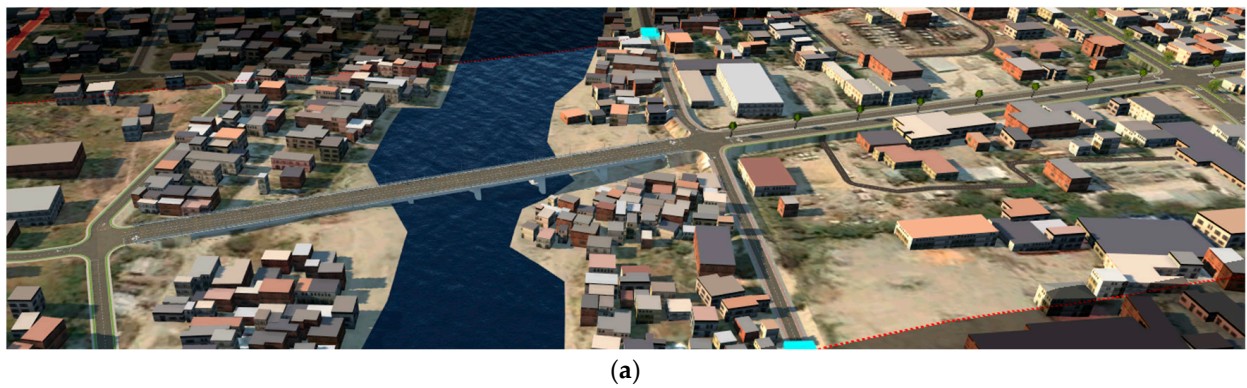

(**a**)

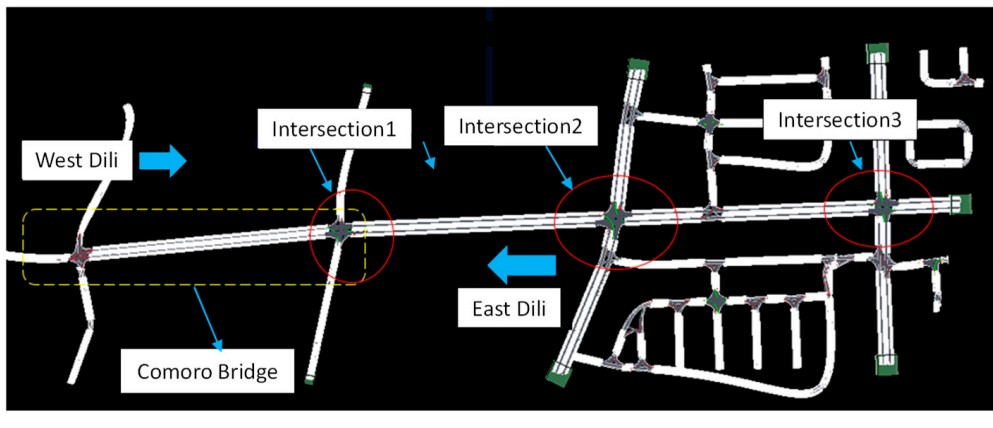

(**b**)

**Figure 12.** *Cont.*

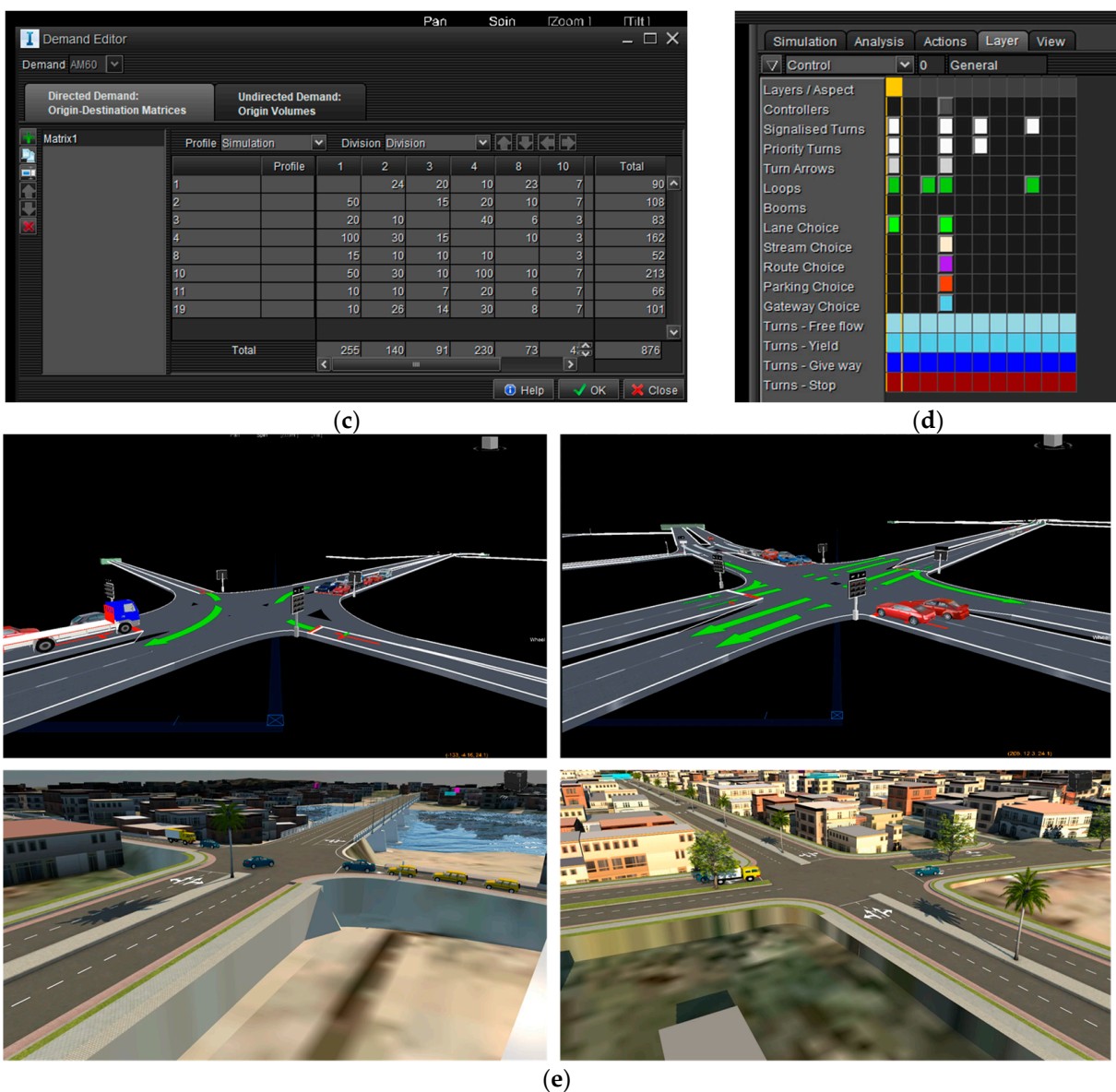

**Figure 12.** Traffic flow simulation in BIM. (**a**) Target area; (**b**) overall area of traffic flow simulation; (**c**) traffic demand panels; (**d**) traffic control panel; (**e**) video simulations.

- To assess the effectiveness of the Comoro Bridge in mitigating traffic congestion in the City of Dili.
- To validate the capabilities of advanced BIM technology in designing and managing transportation infrastructure.
- To demonstrate the efficacy of BIM technology applications in the architecture, engineering, and construction (AEC) industry.

Since the purpose of this paper was to examine how to utilize digital twins for the Comoro Bridge, quantitative analysis results will be presented in another paper. However, preliminary studies predicted that the construction of the Comoro Bridge would alleviate traffic congestion in Dili City. Such traffic flow simulations are extremely useful not only for verifying the impact of the construction of a new bridge but also for predicting traffic congestion on surrounding roads during construction and for considering measures to alleviate traffic congestion at the planning stage.

*4.4. Digital Twin of Comoro Bridge for Structural Performance Evaluation*

4.4.1. Digital Twin–BIM Technology and Finite Element Analysis Interoperability

This study involved the interoperability between digital twin–BIM technology and finite element analysis (FEA), as depicted in Figure 13. The process of interoperability between digital twin–BIM technology and FEA is outlined as follows:

1. The complete Comoro Bridge model was developed using BIM tools.
2. A simplification process was conducted to remove secondary elements such as railings, electrical poles, pedestrians, and other nonessential components.
3. The IFC format was utilized for seamless interoperability between BIM and FE analysis (FEA).
4. After achieving interoperability, the bridge model was prepared for the FE analysis (FEA).
5. The finite element analysis (FEA) was performed.
6. Results and evaluations are conducted.

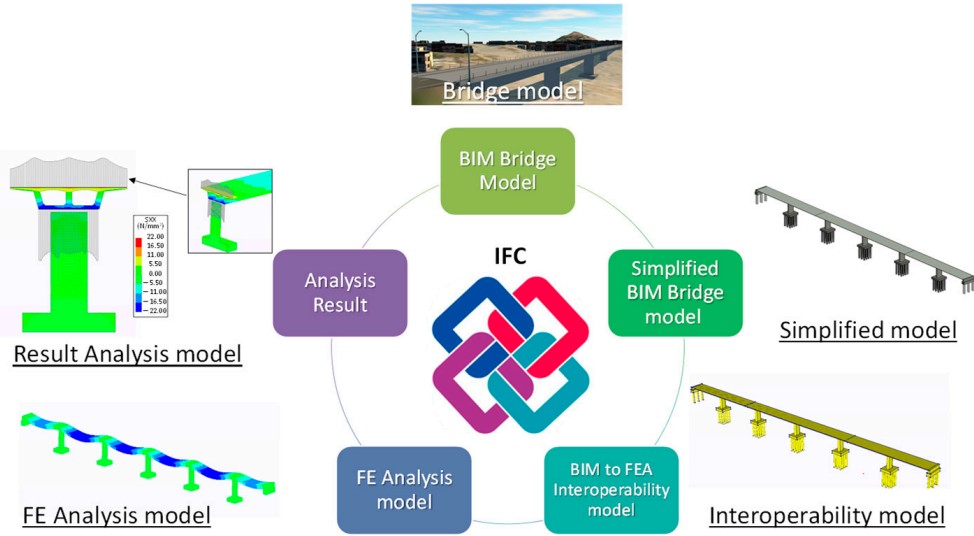

**Figure 13.** Digital twin–BIM and FEA interoperability.

4.4.2. Structural Performance Evaluation Based on FE Analysis

- Full-Scale Model Creation for FE Analysis

The digital twin of the Comoro Bridge, developed in a BIM tool (Revit) and supported by visual programming (refer to Figure 14), facilitated the FE analysis of the full-scale model. Subsequently, the model was exported to IFC files for integration with FE software. Following this, it was necessary to install prestressing tendons along the PC box girder sections based on the construction drawings, as depicted in Figure 15.

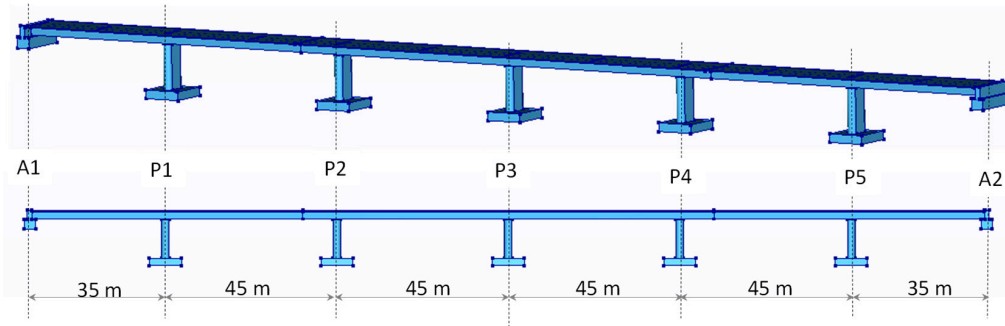

**Figure 14.** Full-Scale Model.

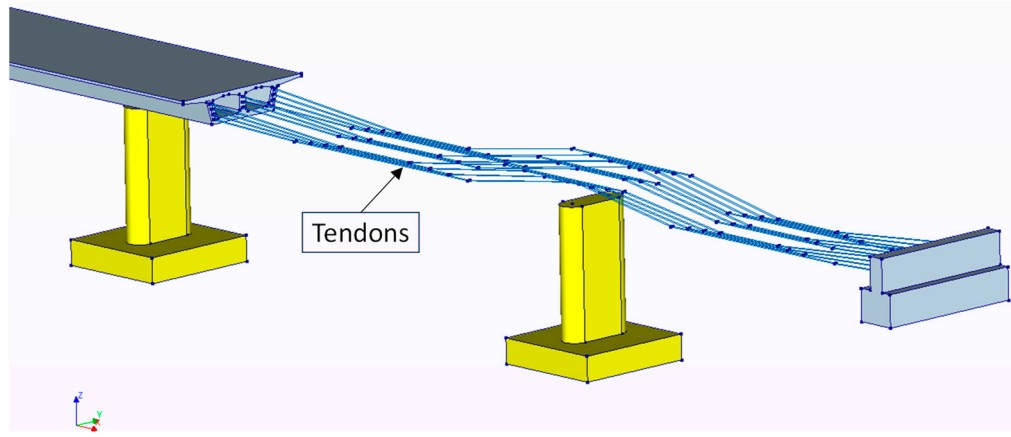

**Figure 15.** Prestressed Tendons Installation.

The Comoro Bridge model was composed of three PC-box superstructures, five piers, and two abutments as mentioned above. Each component was idealized by a solid element, and the bearing on the top of piers and abutments was idealized by spring elements, which represented the stiffness of six degrees of freedom in each direction. The prestressing tendons were idealized by the so-called embedded bar element coupled with the concrete superstructure. The embedded bar element was a virtual element representing the interaction behavior of the reinforcement inside the concrete members.

- Material Properties

The material properties of the concrete and prestressed tendons utilized for the FE analysis in this study are presented in Tables 1 and 2 based on construction records, respectively.

**Table 1.** Material properties of the concrete in the analysis.

| No. | Type of Structure | Materials | |
|-----|-------------------|-----------|--|
| 1 | Superstructure | Concrete (fc) | 36 MPa |
| | | Poisson's ratio | 0.2 |
| | | Mass density | 2500 kg/m$^3$ |
| 2 | Substructure | Concrete | 30 MPa |
| | | Poisson's ratio | 0.2 |
| | | Mass density | 2500 kg/m$^3$ |

**Table 2.** Prestressed tendons.

| | | Main Tendon |
|--|--|-------------|
| Type | | SWPR7BL 12S15.2mm |
| Tensile stress | | 1850 (N/mm$^2$) |
| Yield strength | | 1600 (N/mm$^2$) |
| Allowable tensile stress | At the time of prestressing | 1440 (N/mm$^2$) |
| | Just after prestressing | 1295 (N/mm$^2$) |
| | Design load | 1110 (N/mm$^2$) |

- Boundary Conditions
  The boundary conditions applied in this analysis were as follows:

  ➢ The bottoms of pile caps and abutments were idealized as fixed supports. They were rigidly connected to the ground with translations (T1, T2, and T3) constrained, as illustrated in Figure 16.

  ➢ The bearings were represented by spring materials, as depicted in Figure 17. Modeling the rotational boundary conditions as fully free poses challenges, hence rotational springs were adopted with a stiffness value of $10^6$ (kNm/rad) as indicated in Table 3.

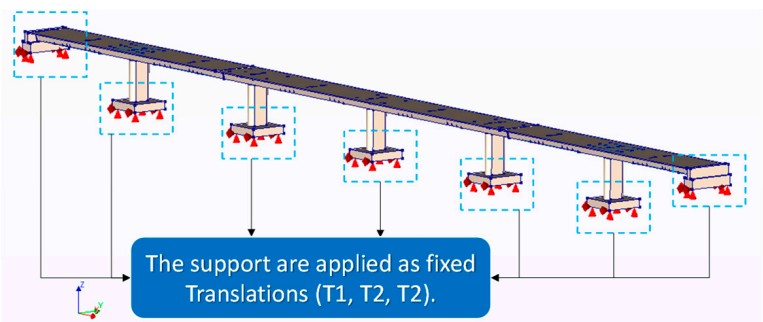

**Figure 16.** Support.

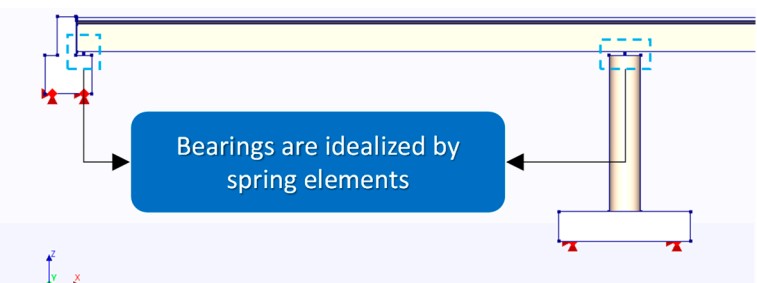

**Figure 17.** Springs.

**Table 3.** Boundary conditions idealized as springs.

| Location | Direction | | Theoretical B. C. | Analysis | |
|---|---|---|---|---|---|
| | | | | B. C. | Stiffness |
| A1 | Sway deformation | $X$ | Fix | Rigidly fixed by spring element | $10^{10}$ (kN/m) |
| | | $Y$ | | | |
| | | $Z$ | | | |
| | Rotational deformation | $R_X$ | Fix | Constrained by spring element | $10^6$ (kNm/rad) |
| | | $R_Y$ | Free | | |
| | | $R_Z$ | Fix | | |
| P1–P5 | Sway deformation | $X$ | Fix | Rigidly fixed by spring element | $10^{10}$ (kN/m) |
| | | $Y$ | | | |
| | | $Z$ | | | |
| | Rotational deformation | $R_X$ | Fix | Constrained by spring element | $10^6$ (kNm/rad) |
| | | $R_Y$ | Free | | |
| | | $R_Z$ | Fix | | |

**Table 3.** *Cont.*

| Location | Direction | | Theoretical B. C. | Analysis B. C. | Analysis Stiffness |
|---|---|---|---|---|---|
| A2 | Sway deformation | $X$ | Fix | Rigidly fixed by spring element | $10^{10}$ (kN/m) |
| | | $Y$ | | | |
| | | $Z$ | | | |
| | Rotational deformation | $R_X$ | Fix | Constrained by spring element | $10^6$ (kNm/rad) |
| | | $R_Y$ | Free | | |
| | | $R_Z$ | Fix | | |

- Loads

    The loads considered in this analysis are depicted in Figure 18 as follows:

1. Self-weight (DL): represents the weight of all structural elements of the bridge.
2. Superimposed dead load (SDL): Includes the weight of the steel railing, asphalt concrete pavement, safety fences, and other components. The combined value of this load was 50 kN/m.
3. Prestress Tendon (T): represents the tendons just after prestressing with a stress of 1295 N/mm$^2$.

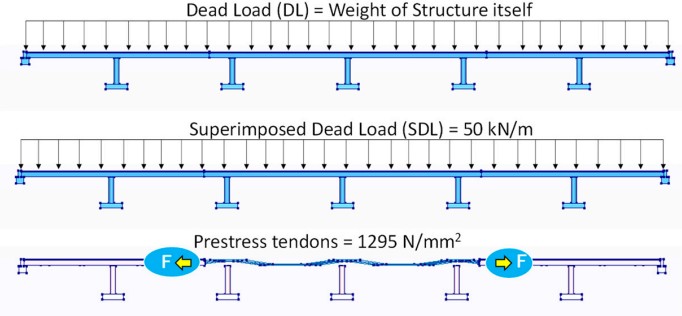

**Figure 18.** Loads.

- Discretization ("Meshing")

    A mesh is a discrete representation of a larger geometric domain using smaller cells. Meshes are commonly employed for solving partial differential equations, rendering computer graphics, and analyzing geographical and cartographic data. Meshing divides the space into elements (or cells or zones) on which the equations can be solved, providing an approximation of the solution across the larger domain. In this study, the meshing process was performed in DIANA FEA, using a mesh size of 0.25 m and a HEX/SQUAD meshing type, as illustrated in Figure 19.

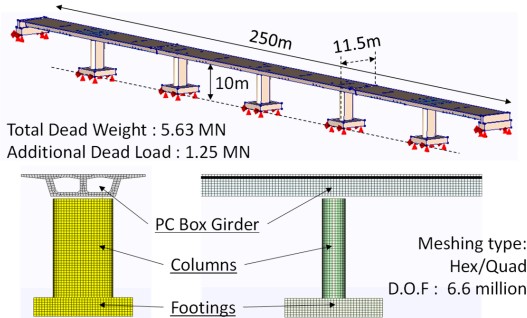

**Figure 19.** Discretization.

- Analysis Type

The types of FE analysis conducted in this study are presented in Table 4. Geometric nonlinear analyses were not performed.

**Table 4.** FE analysis types.

| Names | Types |
|---|---|
| Analysis | Linear static |
| Dimension | Three dimensions |
| Composed element | Solid |
| Mesh order | Linear |
| Mesh size | 0.25 m |
| D.O. F | 6.6 million |

- Structural Evaluation for state conditions of Comoro Bridge

When analyzing the Comoro Bridge model using an FE analysis, the obtained results can be compared to the theoretical values to evaluate the structural performance, as outlined below:

■ Displacement

The displacement results from the dead load (DL + SDL) obtained from both the FE analysis and theoretical calculations are presented in Figure 20. The results indicated that the displacements obtained from both the FE analysis and theoretical calculations were quite similar in the internal spans. However, there was a slight variation in the displacements of the side spans due to the influence of the boundary.

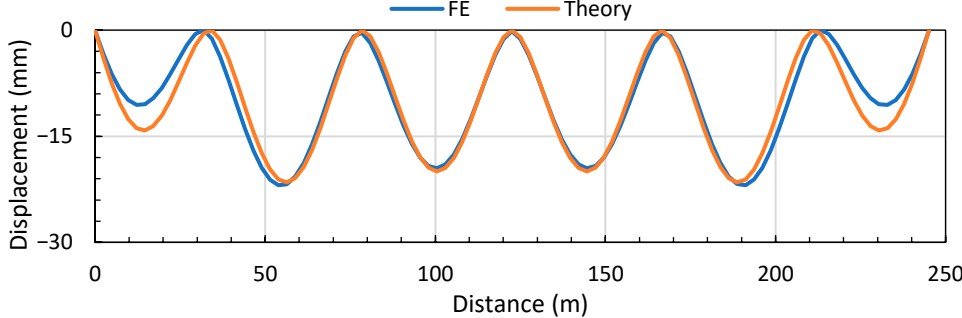

**Figure 20.** Displacement due to the dead load (DL + SDL).

■ Normal stress distribution

The results of the finite element analysis (FEA) can be utilized to assess the distribution of normal stress along the axis of the Comoro Bridge in comparison to the theoretical results. The normal stress distribution presented in this study was induced by the dead load (DL + SDL), as depicted in Figure 21. According to the Euler–Bernoulli theory, the normal stress is distributed according to the bending moment distribution along the bridge axis, resulting in peak values at the top of the piers. As observed in the figure, the normal stress distribution results obtained from both the FE analysis and theoretical calculations were consistent and nearly identical. However, the peak stress occurring at the top of the piers exhibited larger values than those predicted by the Euler–Bernoulli theory. The mechanisms underlying these differences are discussed later.

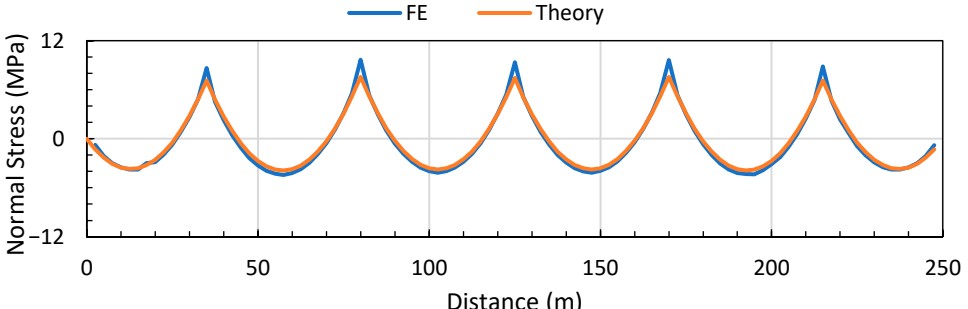

**Figure 21.** Normal stress distributions due to the dead load (DL + SDL).

■     Shear-Stress distribution

Due to the uniform load action of the dead load (DL) and superimposed dead load (SDL), the shear stress exhibited a linear distribution with a downward slope, as illustrated in Figure 22a. Furthermore, the average shear stress, calculated using Equation (1), by dividing the theoretical value of the acting shear force by the equivalent cross-sectional area of the highlighted region in Figure 22b, demonstrated an excellent agreement with the results obtained from the structural analysis. This phenomenon is attributed to the uniform concentration of the shear stress on the web of the prestressed-concrete (PC) box girder, as illustrated in Figure 22c. The detailed FE analysis results obtained using the overall bridge model consistently aligned with the theoretically expected results, providing conclusive evidence of the significant value and usefulness of the digital twin in the bridge analysis.

$$\tau^* = \frac{\textbf{Shear force}(\textbf{Q})}{\textbf{Effective area}(\textbf{A}_{\textbf{eff}})} \tag{1}$$

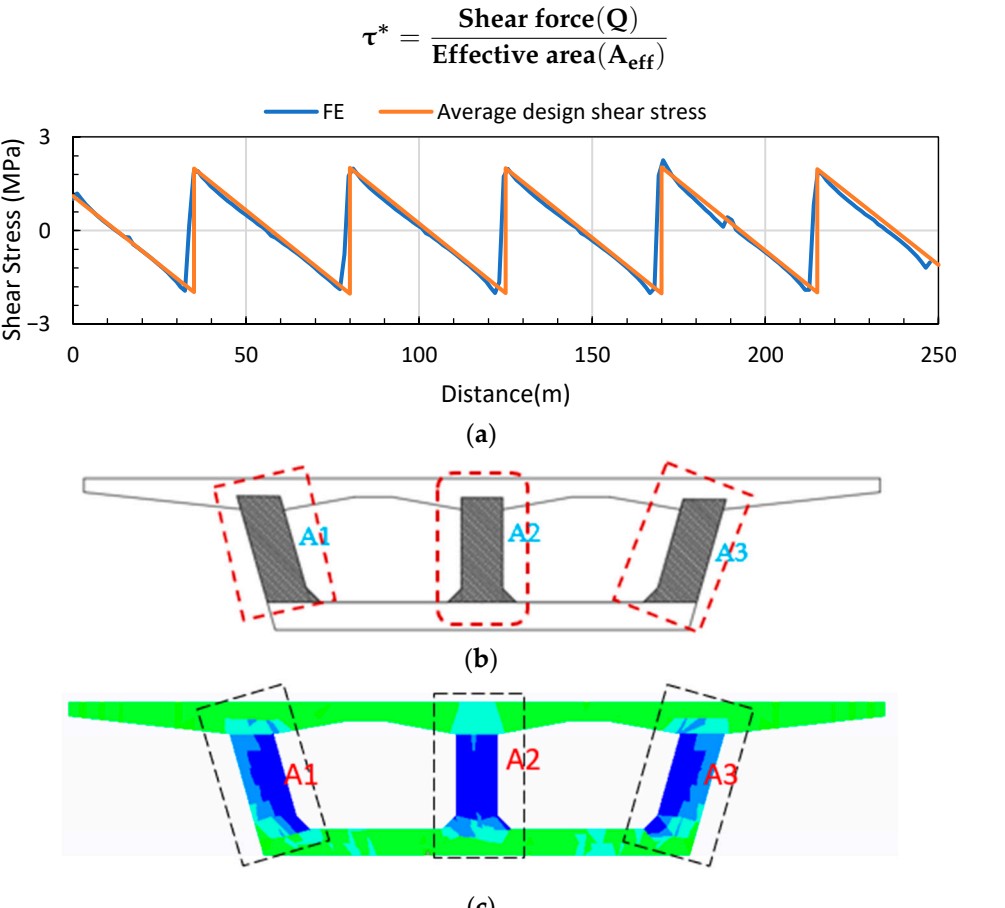

**Figure 22.** Shear-stress distribution along the bridge axis. (**a**) Shear-stress distribution due to the dead load (DL + SDL); (**b**) effective area; (**c**) contour map of the effective area.

- Sectional force (Coefficient Z value)

The finite element analysis enabled the evaluation of the sectional coefficient Z along the bridge axis, obtained using Equation (2), as depicted in Figure 23a. At certain locations, the coefficient Z was arbitrarily distributed; however, taking the average value results in a Z coefficient value of 6.059 m$^3$. This value closely aligned with the theoretical value of 6.195 m$^3$ based on the Euler–Bernoulli theory, as illustrated in Figure 23b.

$$Z = \frac{\text{Inertia Moment}}{\text{neutal axis height}} = \frac{I}{y} \tag{2}$$

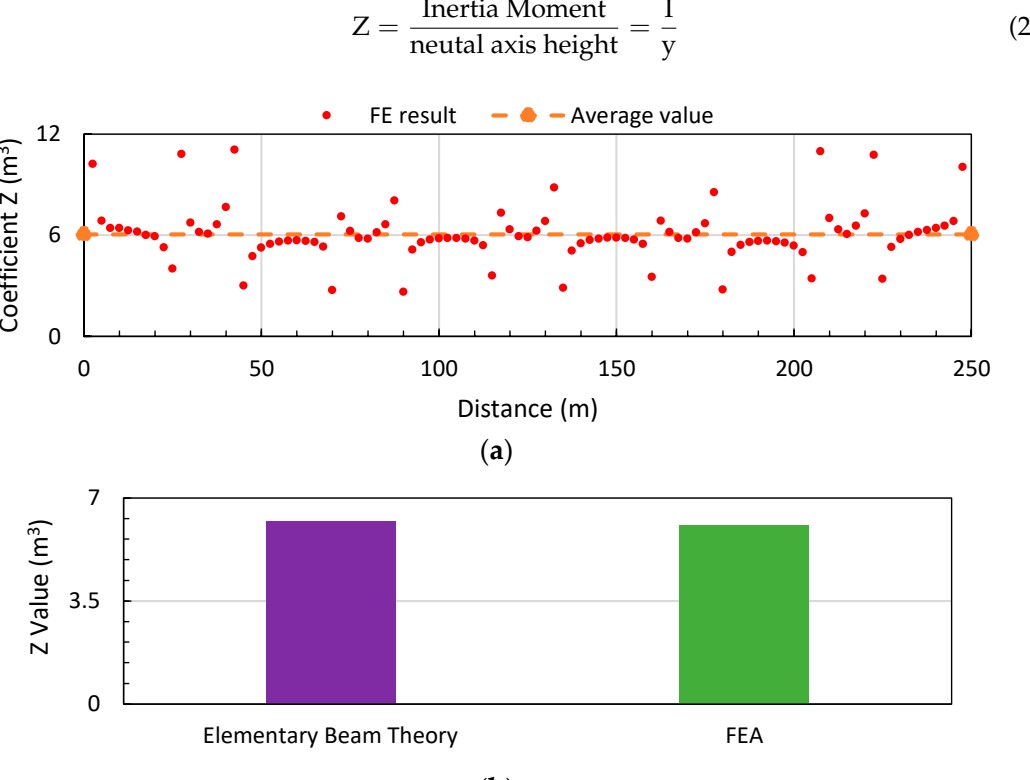

(a)

(b)

**Figure 23.** Coefficient Z value. (**a**) Z value along the bridge axis; (**b**) Z value results from the FEA and theory.

- Nonuniform normal stress (Shear-lag) distribution on box-girder section

The phenomenon of the shear-lag effect can be evaluated in the box-girder section, as depicted in Figure 24a,b. The uniform normal stress distribution along the flange of thin-walled flexural members, such as box girders, is recognized as shear-lag effects [23,24]. According to elementary beam theory (Bernoulli–Euler theory), the normal stress in the longitudinal direction resulting from a bending deformation is assumed to be proportional to the distance from the neutral axis. As a result, it is expected to be uniform across the flange width, as formulated in Equation (3).

$$\sigma_x = \frac{M_y}{I_z}y = \frac{M}{z} \tag{3}$$

However, when the flange width increases, this assumption becomes invalid. The phenomenon of shear-lag effect can also be defined as the nonuniform distribution of normal stress in the wide flange. Nevertheless, the stress reaches its maximum value at the intersection of the flange and web, gradually decreasing towards the center of the flange [23,24].

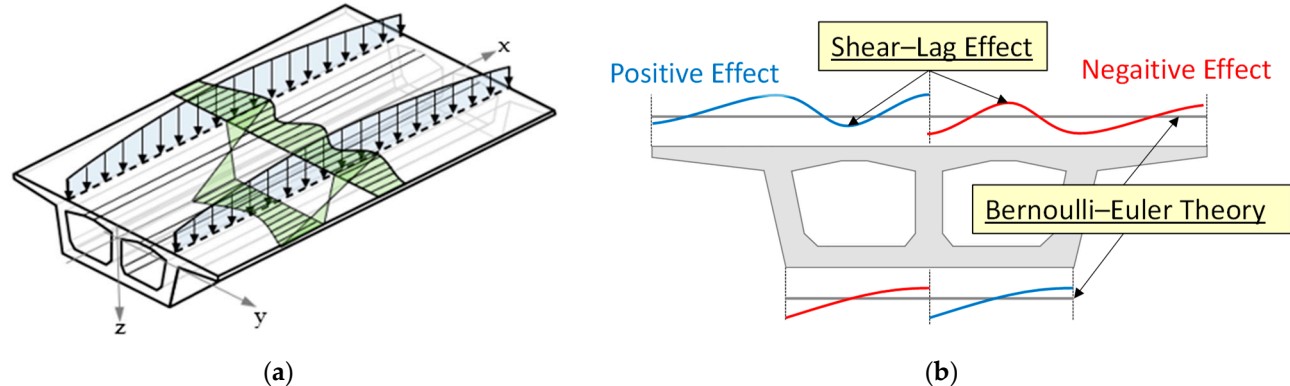

**(a)**            **(b)**

**Figure 24.** Nonuniform normal stress distribution on box-girder section. (**a**) the distribution of nonuniform normal stress along the bridge axis. (**b**) Nonuniform normal stress distribution in the PC box-girder section.

■  Shear-Lag Effect Evaluation on PC Box-Girder Section

To evaluate the shear-lag effect of the PC box-girder section, it is necessary to identify the target locations along the bridge axis. In accordance with Figure 25, the bridge span between pier 3 and pier 4 was selected for evaluation. The chosen evaluation locations included the top piers, the quarter-point along the span length, and the midspan.

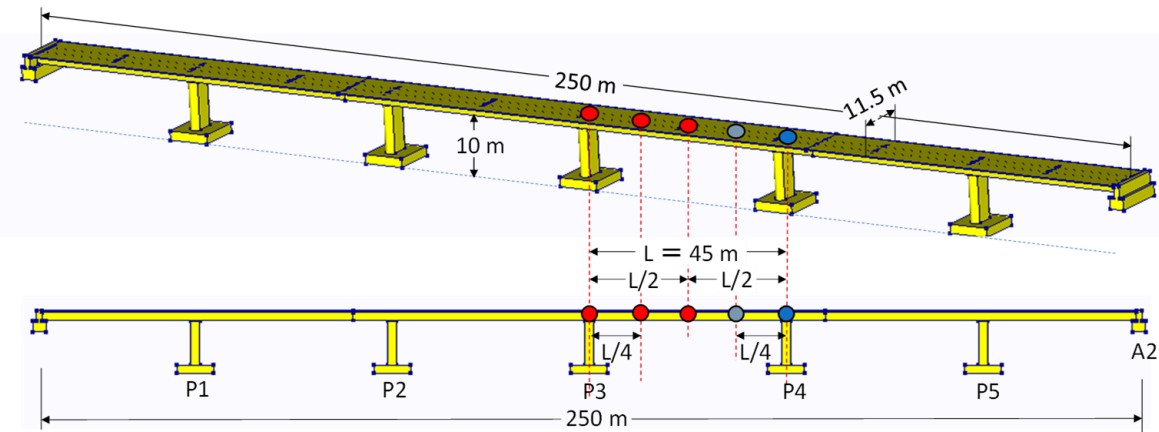

**Figure 25.** Evaluation Point of Shear-Lag Effect.

■  Shear-lag effect Evaluation

The normal stress distribution at the top and bottom flanges of the box girder at pier 3 was evaluated and is depicted in Figure 26a. Figure 26b,c illustrate the occurrence of a nonuniform stress distribution at the top and bottom surface flanges, deviating from the predictions of the Euler–Bernoulli theory. Consequently, a significant shear-lag effect phenomenon was observed on the box-girder section at both the top and bottom flanges at pier 3.

The normal stress distribution at the top and bottom flanges of the box girder, positioned 1/4 L away from pier 3, is depicted in Figure 27a. Figure 27b,c demonstrate the occurrence of a nonuniform stress distribution at the top and bottom surface flanges, deviating from the predictions of the Euler–Bernoulli theory. Nevertheless, the shear-lag effect was notably diminished towards the midspan when considering the 1/4 L location.

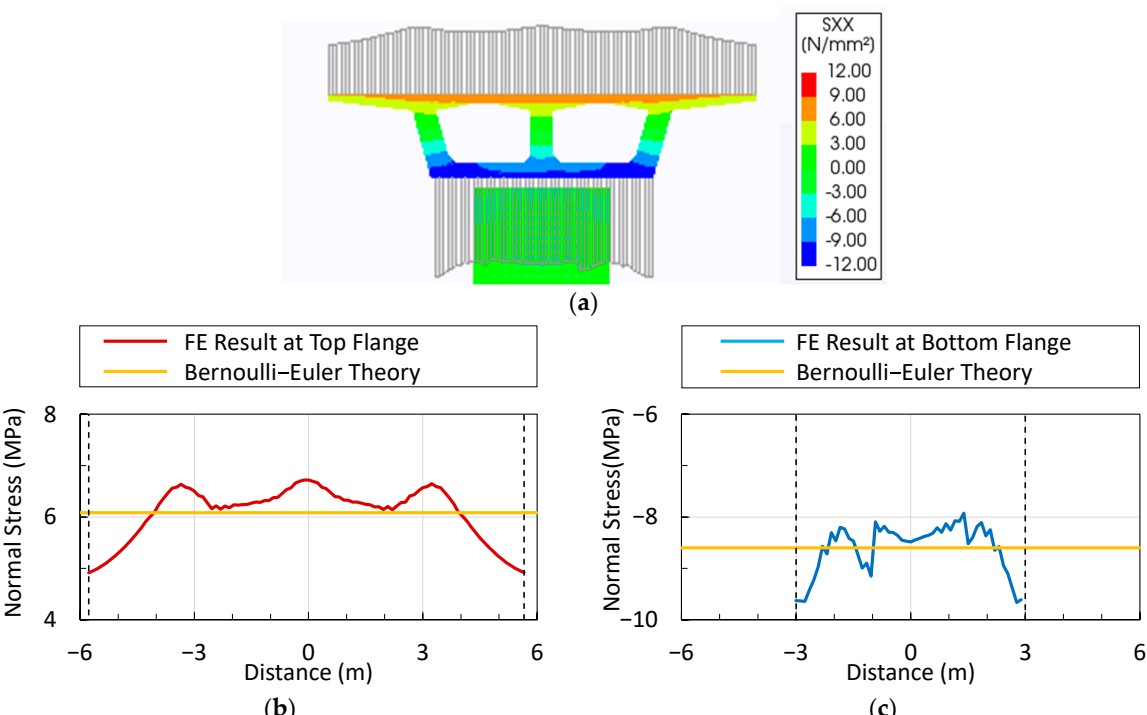

**Figure 26.** Normal stress distribution on the girder section at pier 3. (**a**) Contour map of the normal stress distribution; (**b**) at the top flange; (**c**) at the bottom flange.

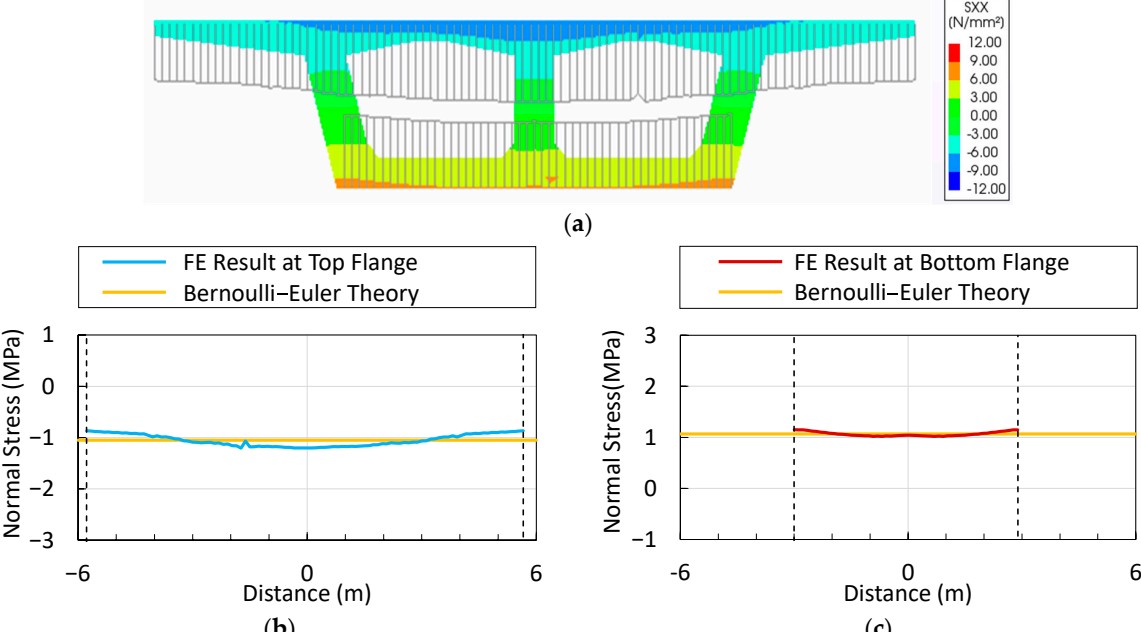

**Figure 27.** Normal stress distribution on the girder section at 1/4 L from pier 3. (**a**) Contour map of the normal stress distribution; (**b**) at the top flange; (**c**) at the bottom flange.

The contour map of the stress distribution (Figure 28a) clearly illustrates that the shear-lag effects at both the bottom and top flanges of the midspan box girders were negligible, closely aligning with the assumptions of the Euler–Bernoulli beam theory (refer to Figure 28b,c). Consequently, given the limited occurrence of shear forces, the shear-lag effect could be disregarded in the midspan section of the box girder.

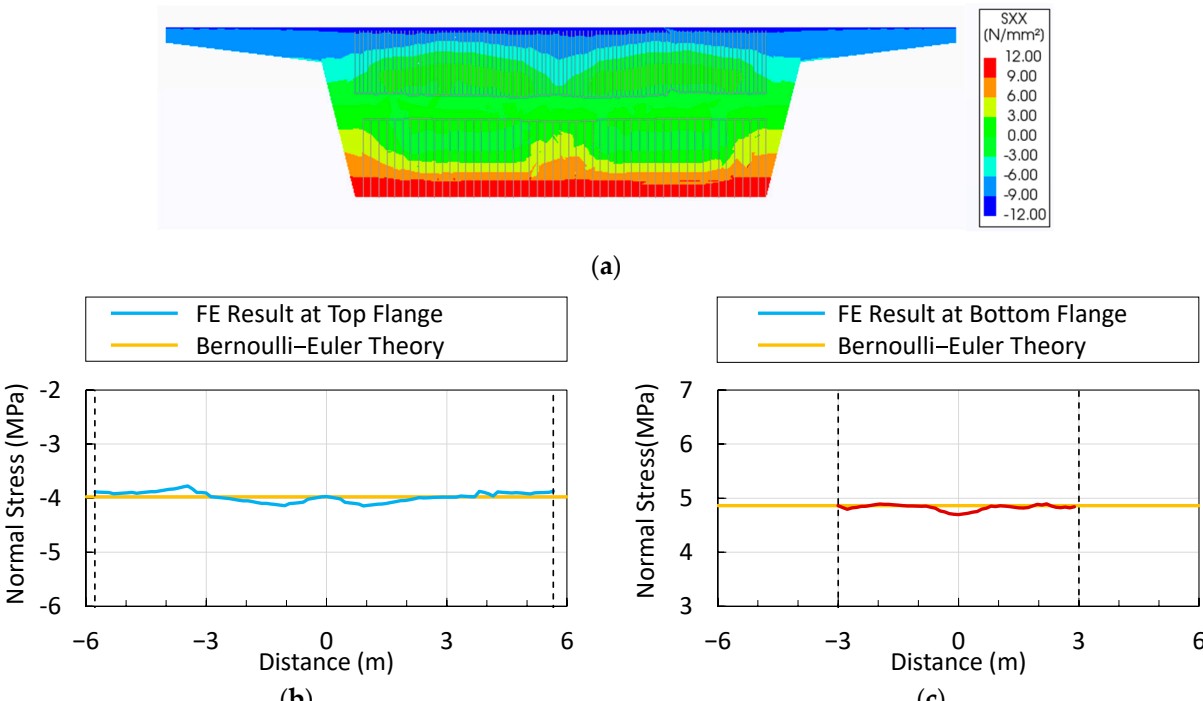

**Figure 28.** Normal stress distribution on the girder section at midspan. (**a**) Contour map of the normal stress distribution; (**b**) at the top flange; (**c**) at the bottom flange.

Therefore, the assessment of the shear-lag effect for the PC box-girder section of the Comoro Bridge was as follows:

1. The nonuniform normal stress distribution due to the shear-lag effect obviously occurred at the top and bottom flanges.
2. The shear-lag effect significantly occurred at the top of the piers, especially at the web and flanges interactions.
3. However, when it moved towards the midspan, the shear-lag effect was diminished linearly and could even be neglected.

- Shear-Lag Effect at Double-Cell Box Vs. Single-Cell Box Girder

In order to investigate the shear-lag effect phenomenon on the PC box-girder bridge in Comoro, a double-cell box girder was designed to replicate the original configuration of the Comoro Bridge compared to a single-cell box girder. The single-cell box girder possessed identical dimensions and cross-sectional area. In the double-cell box girder, the middle web was removed and added to the left and right webs of the single cell-box girder, as illustrated in Figure 29. Consequently, the evaluation of shear-lag effects in both the double-cell and single-cell box girders was as follows:

- The shear-lag effect was observable at the top and bottom flanges in the box-girder section located at the top of the piers, as depicted in Figure 30a.
- The results indicated that the shear-lag effect was more prominent in the single-cell box compared to the double-cell box. This disparity was attributable to the geometrical differences, with the double-cell box featuring an intermediate web, while the single-cell box lacked one (refer to Figure 30b,c).

Consequently, the shear-lag effect was significantly more pronounced in the single-cell box girder. However, as the location transitions towards the midspan, the shear-lag effect diminished linearly and became aligned with the assumptions of the Euler–Bernoulli theory (refer to Figure 31).

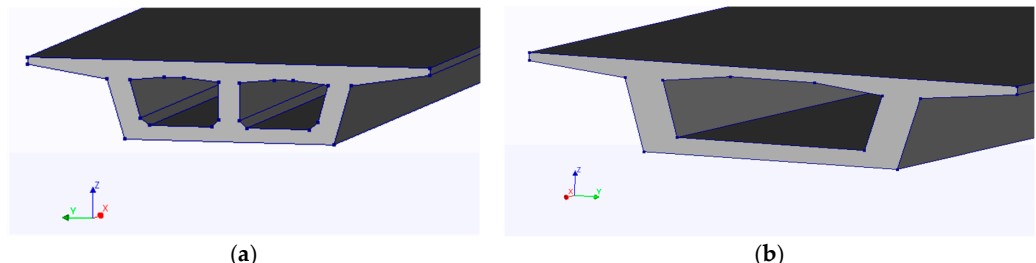

**Figure 29.** Double-Cell Box and Single-Cell Box Girder. (**a**) Double-Cell Box Girder; (**b**) Single-Cell Box Girder.

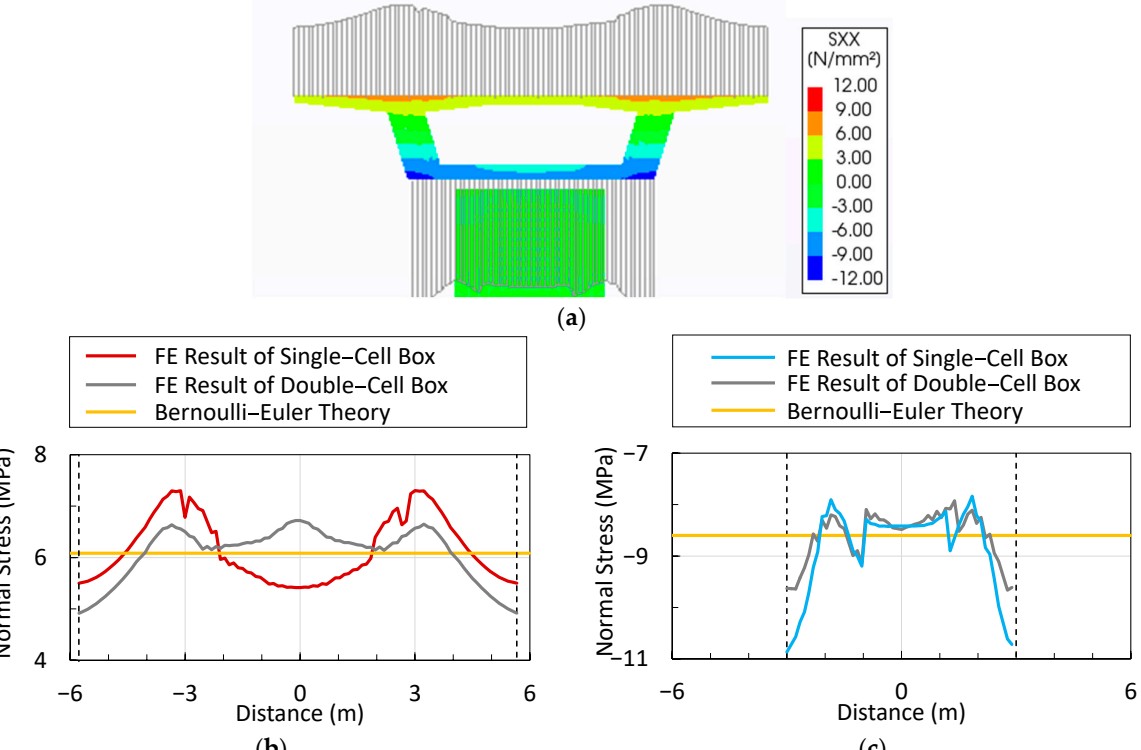

**Figure 30.** Normal stress distribution on the girder section at pier 3 (single-cell box-girder case). (**a**) Contour map of the normal stress distribution; (**b**) at the top flange; (**c**) at the bottom flange.

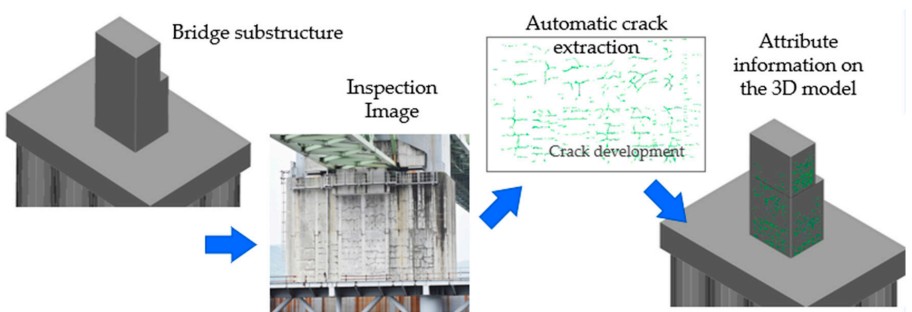

**Figure 31.** Bridge digital twin for structure maintenance.

The evaluation results of the shear-lag effect for both the double-cell box and single-cell box were as follows:

1.  Both the double-cell and single-cell boxes exhibited a shear-lag effect, especially at the top of the piers.

2.　As the location progressed towards the midspan, the shear-lag effect diminished linearly and became aligned with the assumptions of the Euler–Bernoulli beam theory.
3.　At the midspan in the case of the double-cell box, the shear-lag effect was minimal and closely resembled the behavior predicted by beam theory. However, in the single-cell box, where no middle web was installed, the shear-lag effect slightly increased compared to that of the double-cell box.

A comparative study can be performed using a full-scale bridge model, which is easily developed from the BIM technology.

### 4.5. Application of Bridge Digital Twin for Structure Maintenance

Digital twin–BIM technology can be utilized for the maintenance of bridge structures. As illustrated in Figure 32, it is feasible to automatically extract crack information (maintenance data) from captured image data during inspections and incorporate it as attribute information into the 3D model of the structure. This enables the storage and management of maintenance information using BIM technology. Hence, digital twin–BIM technology plays a vital role in managing the complete life cycle of the bridge. Additionally, as mentioned in the Section 1, structural health monitoring has been planned based on the GNSS data observation as depicted in Figure 32. This figure shows the setup of the monitoring system and observation data in past research. By a continuous 3D displacement monitoring of bridges, it is expected that valuable data will be available for the maintenance of the bridge.

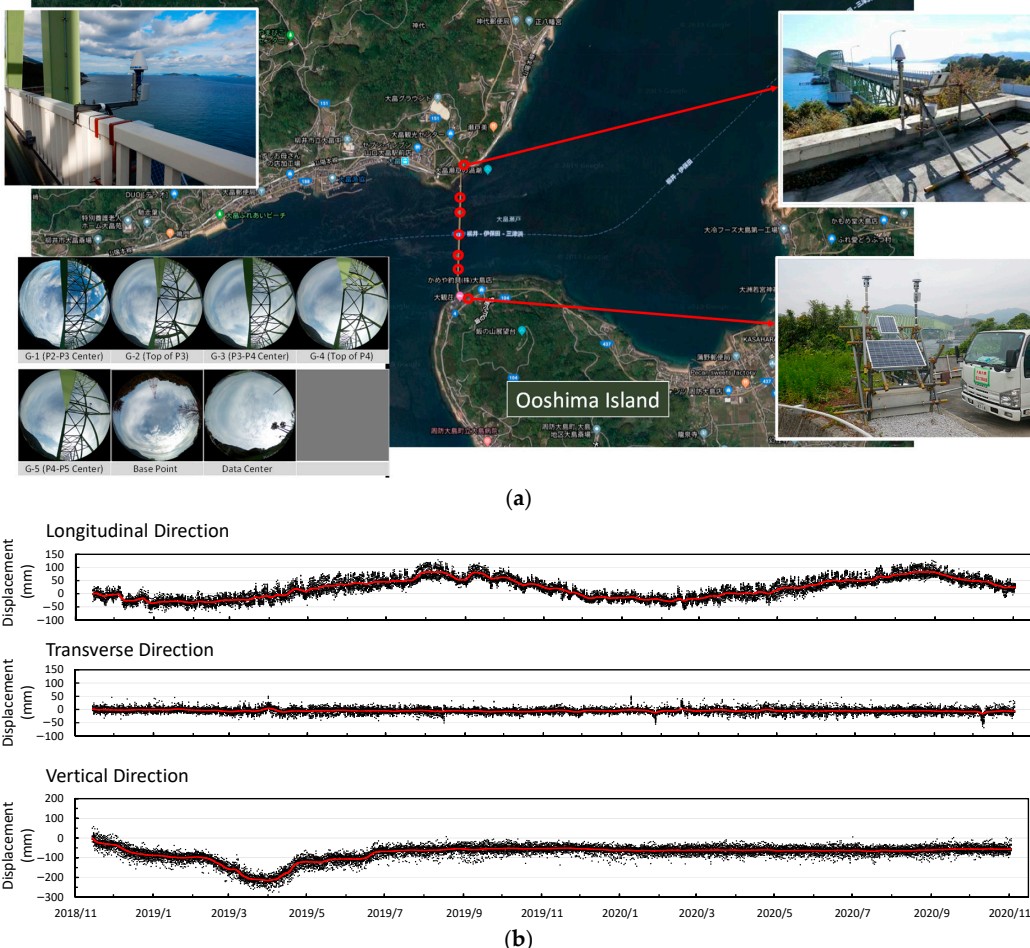

**Figure 32.** Implementation of GNSS monitoring for bridges in past research. (**a**) Setup of GNSS monitoring system; (**b**) observation data from the GNSS monitoring system.

## 5. Conclusions

The results of developing a digital twin of the Comoro bridge based on BIM in this research can be summarized as follows:

- The development and application of digital twin-BIM technology for bridge management offer significant advantages in the field of infrastructure maintenance and operation. By integrating the capabilities of digital twin technology with building information modeling (BIM), bridge managers can create a comprehensive virtual representation of the bridge, capturing its physical and functional attributes.
- Digital twin–BIM technology enables the landscape visualization of the Comoro Bridge with the surrounding environment, traffic flow simulation at the area of the target bridge, a finite element analysis for bridge structure performance evaluation, and bridge structure maintenance. Therefore, digital twin–BIM technology has the potential and ability to access accurate and up-to-date information about the bridge's condition that enhances asset management practices, optimizes maintenance schedules, and extends the lifespan of the structure.
- Additionally, digital twin–BIM technology promotes an effective collaboration among stakeholders involved in bridge management, allowing seamless data exchange, information sharing, and coordinated decision-making. This enhances communication, reduces errors, and streamlines workflows, leading to improved efficiency and cost-effectiveness in bridge maintenance and operation.
- Furthermore, the utilization of digital twin–BIM technology in bridge management contributes to enhanced safety by enabling the evaluation of the structural integrity, the assessment of risk factors, and the implementation of preventive measures. This proactive approach minimizes the likelihood of bridge failures, ensuring the safety of users and surrounding areas.

Overall, the combination of digital twin and BIM technologies presents a powerful tool for bridge management, offering enhanced insights, improved decision-making, and optimized asset performance. As technology continues to advance, it holds enormous potential for revolutionizing the way bridges are designed, built, monitored, and maintained, leading to safer and more efficient infrastructure systems.

**Author Contributions:** Conceptualization, G.W., E.E.T., P.S. and K.A.; methodology, E.E.T. and G.W.; software, E.E.T. and P.S.; validation, G.W. and E.E.T., investigation, E.E.T. and G.W.; resources, G.W. and E.E.T.; data curation, E.E.T. and G.W.; writing—original draft preparation, E.E.T. and G.W.; writing—review and editing, G.W. and E.E.T.; visualization, E.E.T.; supervision, G.W.; funding acquisition, G.W. All authors have read and agreed to the published version of the manuscript.

**Funding:** This research received no external funding.

**Institutional Review Board Statement:** Not applicable.

**Informed Consent Statement:** Not applicable.

**Data Availability Statement:** Not applicable.

**Conflicts of Interest:** The authors declare no conflict of interest.

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
