# Peer review of "Development and Application of Digital Twin–BIM Technology for Bridge Management"

_applsci, doi:10.3390/app13137435_

Round 1

Reviewer 1 Report

The reviewer appreciates the work done by the authors. The contents of the article are interesting and useful information. Nonetheless, the article is NOT well formulated and organized and, in the reviewer’s opinion, the goal of the work must be better explained within Abstract, Introduction and Conclusions. Moreover, the publication in the “Applied Sciences, MDPI” is not recommended unless the following suggestions are taken into account:

1) Introduction. The current state of knowledge relating to the article topic has not been covered and clearly presented, and the authors’ contributions and findings are not emphasized. In this regard, the authors should make their effort to address these issues.

2) Introduction. The following point should be underlined: Bridges require proper management that foresee a scrupulous control of the condition of the structure with time, as well as the execution of the required works for proper maintenance. Furthermore, most of the existing concrete bridges worldwide are now reaching such an age, for which, the degradation phenomena may become a concern. Please refer to the following references to understand the related issues:

-  https://doi.org/10.3141/2220-03

-  https://doi.org/10.1016/j.istruc.2021.10.093

3) Section 2. The geometric and mechanical characteristics of the “Comoro Bridge” have not clearly been illustrated. Please provide more information and introduce figures with schemes of the bridge.

4) The finite element (FE) analyses must be better explained with, particularly, details of the model used. It is not clear how the model of the prestressed concrete bridge is composed (beam elements, plate and shell, etc., with the corresponding amounts and mesh sizes) and how the external loading were applied. Which are the geometric characteristics of the concrete girders ? Which are their boundary conditions ? Moreover, have geometric nonlinear analyses been performed ? Please revise these parts and provide more information about the FE model.

5) Please insert a table/s which list/s the types of FE used, with the corresponding amounts within the FE model and mesh sizes.

6) Section 4. “The tendons just after prestressing with the stress of 1295 N/mm2.” How had this value been determined by the authors ? Please specify.

7) The comparisons between the parameters obtained from the conventional theories and those obtained by FE modeling should also be reported in table/s with the corresponding comparison errors.

8) The further work should be mentioned at the end of the article. Please modify it.

9) Conclusions. The findings must be reformulated based on the analytical and numerical results which have been achieved.

10) The English should carefully be checked and improved. Please also check the grammar and tense errors. It is suggested that the authors should ask a native speaker in English to polish the whole article.

English language requires an extensive editing.

Reviewer 2 Report

The paper studied the application of Digital Twin-BIM technology for bridge management, and choose a PC box girder bridge in Dili City as an example. The paper describe the Twin-BIM method used in traffic management, structural evaluation and bridge maintenance. The concept is very new and the conclusions are prospective, which can be used in similar projects, but some important information are missing in the paper, therefore the paper can be accepted after revisions.

Comments:

1, Why choose the traffic management, structural evaluation and bridge maintenance in the paper, and the structural part take 70% of the contents? There are other aspects of bridge study as well.

2, The traffic study part should compare the actual traffic data with the simulations, which is part of Twin-BIM.

3, The structure analysis part would need some real data for verification, and make sure all the data are feasible to measure on site.

4,The bridge maintenance part is too simple, there are many other things need to be inspect besides the cracks. How to consider it in the system?

No comments about the language.

Reviewer 3 Report

After a thorough review of the paper titled "Development and Application of Digital Twin-BIM for Bridge Management", I regret to inform you that I cannot recommend its acceptance for publication. My main concern with the paper is that it fails to deliver on the promised objective of exploring the potential of digital twin technology in bridge management. While the paper claims to have developed a digital twin based on Building Information Model (BIM) for an existing PC Box Girder Bridge, the research approach adopted in the study lacks the essential elements that characterize digital twin technology. The paper merely discusses the development of a 3D model of a bridge using BIM design tools and visual programming language. While the authors claim that the model is a digital twin, they do not provide any evidence to support this claim. Furthermore, the paper does not explore the unique features and benefits of digital twin technology, such as real-time simulation, predictive analysis, and optimization. The authors do not explain how their model replicates the behavior of the physical bridge in real-time or how it can be used for condition assessment ot prediction. Instead, the paper focuses solely on developing a BIM model for the bridge, which is not a new topic and does not demonstrate the potential of digital twin technology.

Additionally, the paper lacks a clear research methodology and theoretical framework. The authors do not provide a clear explanation of the research approach adopted in the study, nor do they explain the rationale for using the chosen methodology. The theoretical framework is also weak, as the authors do not contextualize their research within the existing literature on digital twin technology and bridge management. The paper fails to deliver on its promise to explore the potential of digital twin technology in bridge management. The research approach adopted in the study lacks the essential elements of digital twin technology, and the paper merely discusses the development of a BIM model for the bridge, which is not a new topic. The lack of a clear research methodology and theoretical framework further weakens the paper's contribution to the field of digital twin technology and bridge management.

There are several grammatical mistakes in the article. The article's English must be improved. 

Round 2

Reviewer 1 Report

The required revisions were carried out and the manuscript can be accepted for publication.

English language requires a moderate editing.
